# Clinical Usefulness of the Inhibitory Control Test (ICT) in the Diagnosis of Minimal Hepatic Encephalopathy

**DOI:** 10.3390/ijerph17103645

**Published:** 2020-05-22

**Authors:** Agnieszka Stawicka, Jerzy Jaroszewicz, Justyna Zbrzeźniak, Natalia Sołowianowicz, Aleksandra Woszczenko, Magdalena Świderska, Robert Flisiak

**Affiliations:** 1Department of Infectious Diseases and Hepatology, Medical University of Bialystok, 15-540 Bialystok, Poland; jerzy.jr@gmail.com (J.J.); j.zbrzezniak@op.pl (J.Z.); n.kilisinska@gmail.com (N.S.); woszczenkoaleksandra@gmail.com (A.W.); magdalena.swiderska@umb.edu.pl (M.Ś.); doctors@umb.edu.pl (R.F.); 2Department of Infectious Diseases and Hepatology, Medical University of Silesia, 40-055 Katowice, Poland

**Keywords:** minimal hepatic encephalopathy, MHE, liver encephalopathy, liver cirrhosis, inhibitory control test, ICT

## Abstract

*Background*: Minimal hepatic encephalopathy (MHE) refers to a number of neuropsychiatric and neurophysiological disorders in patients with cirrhosis who do not show abnormalities on physical examination or in clinical tests. The aim of this study was to determine the prevalence, risk factors, and predictive value of minimal hepatic encephalopathy and the usefulness of the inhibitory control test (ICT) in the diagnosis. *Methods*: Seventy patients (mean age 53 years, range 24−77) with liver cirrhosis were enrolled in the study. MHE was diagnosed based on PHES (psychometric hepatic encephalopathy score) and ICT. PHES and ICT were validated in a group of 56 control subjects. *Results*: Minimal hepatic encephalopathy was diagnosed using PHES in 21 patients (30%). ICT diagnosed MHE in 30 patients (42%), and the test had a sensitivity of 65% and a specificity of 57% compared to PHES. The ICT score (lures/target accuracy rate) correlated with the age of subjects (R = 0.35, *p* = 0.002) and only slightly with education (education in years R = −0.22, *p* = 0.06). MHE diagnosed with PHES or ICT was associated with a significantly higher model of end-stage liver disease (MELD) score in the follow-up. MHE diagnosed with ICT was correlated with a significantly higher incidence of symptoms of decompensated cirrhosis (*p* = 0.02) in the follow-up. *Conclusions*: ICT had moderate sensitivity and specificity in diagnosing MHE compared to PHES. Importantly, MHE detected with PHES or ICT was associated with poorer survival and a more severe progression of the disease.

## 1. Introduction

Minimal hepatic encephalopathy (MHE) is defined as the presence of abnormalities in additional tests or clinical symptoms of cerebral dysfunction in patients with chronic liver disease but without symptoms of overt encephalopathy [1,2,3]. The analysis of a variety of psychometric tests in patients with cirrhosis and chronic liver disease has shown impaired reaction time, visual perception, and concentration, especially associated with focusing attention on one task for a longer time, without any abnormalities in verbal performance [4]. Another study has ruled out the impact of general intelligence, sleep, and consciousness disorders on the results of psychometric tests [5]. Memory impairment is an important symptom of MHE. Disorders concern short-term memory and seem to have no effect on the long-term cognitive processes. Patients show dysfunctions in encoding information but have normal reception (memory retrieval). Significant abnormalities are also seen in emotional life, social interactions, and work [6,7]. All the aforementioned processes have a detrimental effect on the quality of life of patients with MHE. One important consequence of MHE is an impaired fitness to drive and its strong negative effect on the number of motor vehicle crashes and traffic violations [8]. This is due to impaired attention and reaction time, especially in situations requiring prompt action, such as a pedestrian rushing across the road. Bajaj and collaborators in a questionnaire-based survey found that cirrhotic patients versus control subjects more frequently caused road accidents within both one and 5 years (9% vs. 1% (1yr); 17% vs. 4% (5yr)). Patients with cirrhosis also caused more traffic violations (25% vs. 4% (5 yr)), 13% vs. 2% (1 yr)). Multivariate regression analysis showed that the MHE+ status was the only risk factor, increasing the number of road accidents and traffic violations in patients with cirrhosis. The number of traffic violations in the group of MHE+ patients was higher than in MHE– patients within one year (21% vs. 4%; *p* = 0.003) and 5 years (36% vs. 12%; *p* = 0.004), and so was the number of road accidents within one year (17% vs. 0%; *p* = 0.03) and 5 years (33% vs. 12%; *p* = 0.03) [9]. Importantly, MHE may precede the onset of overt hepatic encephalopathy, which indicates the role of MHE as a predictive factor for the progression of cirrhosis. The prevalence of covert hepatic encephalopathy among cirrhotic patients ranges from 20% to 80% [10]. The precise scale of the problem is extremely difficult to identify due to the lack of standardized and widely accepted diagnostic methods. Different results in studied populations have been obtained depending on the adopted method and criteria. The recommended diagnostic methods for MHE include the psychometric hepatic encephalopathy score (PHES), which, despite elapsing time, remains the gold standard in diagnostics, the critical flicker frequency test (CFF), continuous reaction time test (CRT), the inhibitory control test (ICT), the Stroop test in the form of an application for mobile devices, the computerized SCAN test, and the classical EEG examination. Any of these tests is suitable as long as it has been validated on a local population. For research purposes, MHE should be diagnosed based on at least two tests: the obligatory PHES and one of the following computerized tests: CRT, ICT, SCAN, Stroop test, or neurophysiological CFF or EEG [1]. So far, one paper has been published in Poland on the standardization of PHES in a population of 317 healthy volunteers [11]. ICT examines the concentration of attention and response inhibition. Initially, this test was used for the examination of patients with schizophrenia, ADHD, or traumatic brain injury [12,13,14]. Bajaj et al. proposed the use of ICT for the diagnosis of patients with cirrhosis [15]. The aim of this study was to determine the prevalence, risk factors, and predictive value of minimal hepatic encephalopathy, as well as to assess the usefulness of the inhibitory control test in the diagnosis of MHE.

## 2. Materials and Methods

### 2.1. Ethical Considerations

Each patient signed a written informed consent form before inclusion in the study. The study protocol was approved by the Bioethical Committee of the Medical University of Białystok (R-I−002/254/2015) in accordance with the GCP (guidelines for good clinical practice) rules.

### 2.2. Materials and Methods

#### 2.2.1. Study Group

Seventy patients (mean age 53 years, range 24−77, 45 male) with cirrhosis diagnosed based on clinical symptoms were enrolled in the study. Liver failure in patients was related to alcohol (37, 53%), infection with HBV or HCV (16, 23%), or autoimmune diseases (17, 24%). Detailed characteristics of the study population are presented in Table 1. Exclusion criteria included the presence of neurological or psychiatric disorders, alcohol abuse (min. 3 months), psychoactive agents, and overt encephalopathy. Exclusion criteria were the medications, such as narcotics, benzodiazepines, tricyclic antidepressants, antipsychotics, and gabapentinoids. Furthermore, subjects with hepatocellular carcinoma or other malignancies were excluded. The mini-mental state examination (MMSE) was performed for that purpose, and a score below 23 was used as the cut-off value for cognitive impairment, disqualifying a patient from further participation in the study. The test was an independent exclusion criterion. The severity of liver disease was determined by the Child–Pugh score and the MELD score. The ammonia level in blood was additionally measured.

Controls. The PHES test was standardized in a group of 56 healthy volunteers (28 male, median age 54 (26−79) years, median education in years 12 (8−23)). All subjects were able to read and write and had no motor or visual impairment. Exclusion criteria included liver disease, neurological diseases, use of psychoactive agents, and alcohol abuse in the three months before the study. Control subjects underwent ICT testing.

#### 2.2.2. Methods

PHES test. PHES, containing 5 psychometric paper-pencil tests, was performed in the study population: NCT-A (number connection tests A) and NCT-B (number connection tests B), SDT (serial dotting test) in which subjects need to draw a dot in the central point of each of 100 circles, DST (digit-symbol test) in which subjects need to transcribe symbols accurately and quickly corresponding to numbers in a timed manner over 90 s, and LTT (line drawing test) in which subjects need to draw a line between two lines on the paper and stay between, neither touching nor drawing over the printed lines [16,17]. Before taking the test, subjects were given instructions presented with an example key to the task and the correct way to solve it. The assessment was carried out in two quiet rooms under similar lighting conditions. Subjects were usually tested between 15:00 and 18:00 and supervised by suitably trained personnel. The scoring system was consistent with the generally adopted principles [18,19]. Scores obtained by control subjects were adjusted to eliminate the impact of age and education in years. Depending on the value of standard deviation, each test done in the control group was scored, respectively, from −3 to +1 for the tests in the study group. The final result was the sum of scores for individual tests in the range of +5 to −15. Scores below −5 were the cut-off value for the diagnosis of MHE [11,20,21].

ICT. ICT (inhibitory control test) is a computerized test of neuronal inhibition. Testing was carried out using free online software (https://www.chronicliverdisease.org/disease_focus/ICT/) developed by Bajaj et al. [15]. The test consists of the presentation on a computer screen of several letters at 500-ms intervals. Subjects are instructed to respond to a specific sequence of letters X and Y, but only when X follows Y or Y follows X, and refrain from responding when X follows X or Y follows Y. The test run consists of 1728 random letters in between, 212 *targets* (alternating X and Y) and 40 *lures* (X and Y not alternating). The whole test consists of one training run and 6 test runs, which are about 2 min. long. At the end of the test, the numbers and rates of lure and target responses are automatically calculated [22]. ICT was performed by 70 cirrhotic patients and 56 control subjects.

#### 2.2.3. Statistical Analysis

Data are presented as mean ± standard deviation (SD) or range. The analysis was performed with the nonparametric Mann–Whitney U and Pearson Chi-square or Fisher exact tests for group comparison and the Spearman correlation test. Statistical significance was adopted at the level of *p* < 0.05. The diagnostic usefulness of ICT in relation to PHES was assessed based on the analysis of receiver operating characteristic (ROC) curves and the area under the curve (AUC). Statistics were processed using two types of software: Statistica 11.0 (Statsoft, Tulsa, OK, USA) for statistical analyses and GraphPad Prism 5.0 (GraphPad, Inc., La Jolla, CA, USA) for graphs.

## 3. Results

The study was performed from March 2014 to January 2016. Minimal hepatic encephalopathy was diagnosed using PHES in 21 patients (30%). The prevalence of MHE was significantly higher in older patients (mean age ± 58.8 vs. 51.3 ± years, *p* = 0.01) and cirrhosis related to alcohol (76% vs. 43%, *p* = 0.01). In addition, the prevalence of diabetes type 2 in MHE+ patients was also significantly higher compared to controls (48% vs. 16%, *p* = 0.006). Patients diagnosed with MHE had higher baseline levels of creatinine (0.9± vs. 0.7± mg/dL, *p* = 0.02). Importantly, there were no significant differences between MHE+ and MHE– patients in relation to the levels of bilirubin, albumin, creatinine, or ammonia (Table 1). Ascites were more frequent in MHE+ patients compared to MHE– patients (71% vs. 45%, *p* = 0.04).

ICT was completed by all subjects. Patients diagnosed with MHE based on PHES made significantly more incorrect responses when assessed with ICT. The rate of correct responses in MHE+ patients was 84.7 ± 12.7% vs. 92.1 ± 7.4%, *p* = 0.01, and a similar relationship was found for the lures/target accuracy rate in ICT (30.6 ± 15.7 vs. 21.3 ± 14.7, *p* = 0.01). ROC analysis revealed that both the percentage of correct responses and the lures/target accuracy rate in ICT were useful for the diagnosis of MHE. AUC for the rate of correct responses and the lures/target accuracy rate was 0.69, *p* = 0.01 (Figure 1). The optimal cut-off for the rate of correct answers in ICT used as a diagnostic tool for MHE was 90.3%, providing a sensitivity of 65% and a specificity of 57%. The cut-off for the lures/target accuracy rate was 23.1, with a sensitivity of 70% and specificity of 67%. ICT (lures/target accuracy) detected MHE in 30 patients (42%). Of note is that ICT score (lures/target accuracy) correlated with the age of subjects (R = 0.35, *p* = 0.002) but not with education (education in years R= −0.22, *p* = 0.06).

For 54 patients, follow-up data were available concerning the monitored MELD score, as well as episodes of decompensation (development of overt encephalopathy, ascites, variceal bleeding) and death. The mean follow-up was 17 months (range 3−37 months). During the follow-up, the median MELD score increased in MHE+ patients diagnosed with PHES by 4.1 points (*p* < 0.001) and in MHE– patients by median 3.0 points (*p* < 0.01, Figure 2). The MELD score also increased in the follow-up on average by median 3.4 points (*p* < 0.005) in MHE+ patients diagnosed based on ICT and by median 2.4 points (*p* = 0.06) in MHE– patients. The diagnosis of MHE based on ICT was significantly correlated with the history of decompensation symptoms (*p* = 0.02), but no such correlation was found for MHE diagnosed with PHES (*p* = 0.13). Assessment with ICT revealed decompensated liver function in only 3 out of 28 non-MHE patients (11%), while 89% of them had stable disease. In contrast to that, MHE undetected with PHES less frequently indicated disease progression without decompensation (82%), while symptoms of liver failure occurred in 7 out of 38 patients in this group (18.4%). Table 2 presents data on the episodes of decompensation and no disease progression based on the previous diagnosis of MHE with PHES or ICT in the study group.

## 4. Discussion

The significant role of minimal hepatic encephalopathy confirmed by clinical data indicates the need for simple, effective, widely available, and reliable diagnostic methods. The imperfection of existing methods discourages researchers from diagnosing MHE. Some studies have demonstrated that patients with MHE significantly improve driving simulator performance and quality of life after treatment with rifaximin or lactulose [23,24]. In a Polish study published in 2013, Wunsch et al. detected MHE with the PHES test in 22% of patients. In our study, minimal hepatic encephalopathy was diagnosed using PHES in 30% of patients [11]. The analyzed inhibitory control test has many advantages: it is widely available, free of charge, does not need preparation on the part of the investigator, and the results are calculated and averaged automatically. ICT has good test-retest reliability. Despite the obvious advantages of this test, such as simple performance (only one key—the space bar—needs to be pressed), in the studied population of mean age 54 years, one limiting factor was that before taking the test, subjects reported concerns about their lack of computer skills or impaired vision. Obtained results indicated moderate sensitivity (65%) and specificity (57%) of ICT compared to PHES.

Recently reported data on the predictive value of ICT are inconsistent. For example, one study carried out in India on 200 patients with cirrhosis and 200 control subjects found that MHE was diagnosed using ICT in 135 patients (67.5%). Mean ICT lures were higher in cirrhotic patients with MHE than those without MHE (17.27, 95% CI 13.9−22.3 vs. 8.79, 95% CI 6.8−12.60, *p* < 0.001) or controls (8.47, 95% CI 6.3−12.3, *p* < 0.05). ICT had a sensitivity of 92.6% and a specificity of 78.5% compared to PHES. Interestingly, ICT, when used for the diagnosis of MHE, correlated with the Child–Turcotte–Pugh class (*p* < 0.001) and MELD score (*p* < 0.001) and predicted the development of overt hepatic encephalopathy in the follow-up (mean 357 days) [21]. In another study published in 2012 and also carried out on an Indian population (a group of 100 patients), ICT had a slightly lower sensitivity (78%) and specificity (65.6%) than PHES. Similarly to our study, Taneja et al. reported that in patients with cirrhosis, ICT did not correlate with the severity of liver disease measured by the Child–Turcotte–Pugh score (r = 0.044, *p* = 0.658) and MELD score (r = 0.176, *p* = 0.077). No follow-up data on patients were assessed in this study [20]. Another study published in 2016 in the USA covered several clinical centers and assessed one of the latest methods for MHE diagnosis—EncephalApp for mobile devices. Investigators used the diagnostic standards of PHES and ICT to validate the usefulness of EncephalApp. The study covered 437 cirrhotic patients and 308 controls. All subjects underwent three diagnostic tests, which yielded different scores. Based on data for controls, MHE was detected in 37% of cirrhotic patients using PHES, 35% using ICT, and in 51% MHE using EncephalApp. Using PHES as the gold standard (total score > −4), EncephalApp detected MHE in 37% of patients (sensitivity 80%). EncephalApp, based on ICT (number of lures), diagnosed MHE in 54% of patients (sensitivity 70%). In the follow-up, the development of overt hepatic encephalopathy was predicted in 13% of patients. EncephalApp as the gold standard for MHE diagnosis was a significantly better predictor of the development of overt hepatic encephalopathy (OHE) (HR 2.1, *p* = 0.04) and increased MELD score (HR 1.4, *p* = 0.05). When ICT was used as a norm for EncephalApp, the time to onset of OHE was dependent on the MELD score. The EncephalApp test diagnosed MHE regardless of cut-off points adopted in comparison with other tests or norms established for the control and was a good independent predictor for the development of OHE [25]. It is important to mention that in our study, the cut-off for abnormal PHES had been set as ≤ −5 SD, which is a more conservative approach, but naturally, the clinical usefulness of ICT results might differ significantly compared to studies in which the PHES cut off point was ≤ −4.

Our study carried out on a Polish population also demonstrated the significant predictive capability of PHES or ICT in MHE diagnosis in patients in which PHES or ICT diagnosed MHE showed a statistically greater increase in MELD scores in the follow-up. Interestingly, an episode of decompensation (overt hepatic encephalopathy, ascites, variceal bleeding) in the follow-up occurred significantly more frequently only in MHE patients diagnosed with ICT. It seems, therefore, that the assessment of MHE with PHES or ICT in all cirrhotic patients is an important predictive factor. The assessment method, however, should be chosen depending on the experience of the investigator and patient preferences. Apparently, older patients tend to prefer the PHES test, but on the other hand, the performance of the ICT test is easier from the point of view of the investigator. Regardless, the study possessed some limitations, among the relatively small number of participants, especially in MHE+ groups, but also a quite significant proportion of patients lost to follow-up.

## 5. Conclusions

To summarize, ICT had moderate sensitivity and specificity in diagnosing MHE compared to PHES. Importantly, MHE diagnosed with PHES or ICT seemed to be associated with poorer survival and a more severe progression of the disease. Patients with MHE had more serious impairment of liver function expressed in the MELD score in the follow-up compared to patients without MHE, which justifies testing all patients with cirrhosis.

## Figures and Tables

**Figure 1 ijerph-17-03645-f001:**
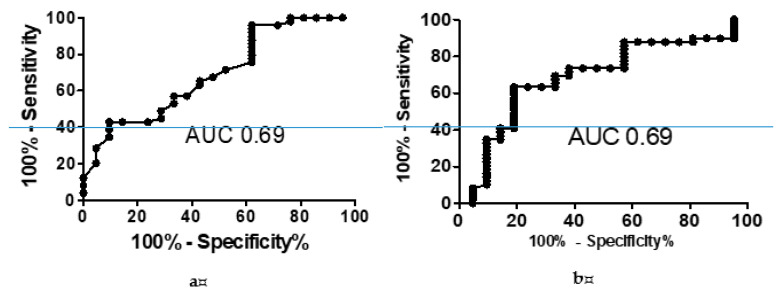
ROC (receiver operating characteristic) curve and AUC (area under the curve) value for ICT (inhibitory control test) in the diagnosis of MHE (minimal hepatic encephalopathy). (**a**) for the rate of incorrect responses, (**b**) for the lures/target accuracy rate.

**Figure 2 ijerph-17-03645-f002:**
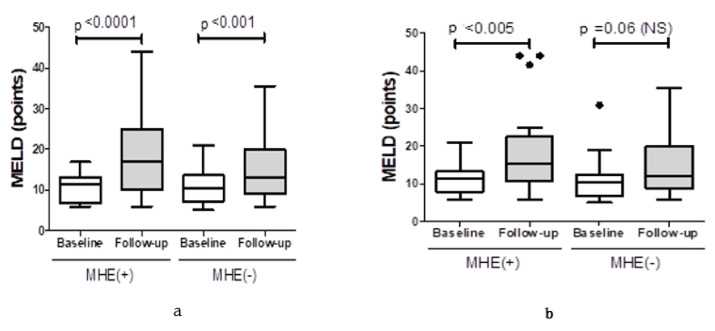
Comparison of baseline and follow-up MELD scores for end-stage liver disease in patients diagnosed with MHE based on psychometric hepatic encephalopathy score (PHES) (**a**) and ICT (**b**).

**Table 1 ijerph-17-03645-t001:** Characteristics of patients with MHE diagnosed based on PHES.

	MHE+ (*n* = 21)	MHE– (*n* = 49)	*p*
Sex (male; *n*, %)	15 (71%)	30 (61%)	0.41
Age (mean, SD)	58.8 ± 11.2	51.3 ± 10.0	0.01 *
Etiology of cirrhosis (*n*, %) ALD HBV/HCV Others	16 (76%) 1 (5%) 4 (19%)	21 (43%) 15 (27%) 13 (26.5%)	0.09
Diabetes (*n*, %)	10 (48%)	8 (16%)	0.006 *
Hypertension (*n*, %)	11(52%)	20 (41%)	0.37
Bilirubin (mg/dL)	4.7 ± 8.2	3.0 ± 4.5	0.46
Albumins (g/dL)	3.1 ± 0,6	3.4 ± 0.6	0.1
INR	1.3 ± 0.3	1.2 ± 0.3	0.06
Creatinine (mg/dL)	0.9 ± 0.4	0.7 ± 0.3	0.02 *
Child-Pugh score	7.9 ± 1.4	7.3 ± 1.8	0.21
MELD score	11.8 ± 7.0	10.6 ± 4.4	0.52
Ammonia (µm/L)	141 ± 51	131 ± 38	0.96

* *p*-value.

**Table 2 ijerph-17-03645-t002:** Progression of disease depending on the previous diagnosis of MHE based on PHES and ICT. *p*-values obtained by Fisher exact test for PHES = 0.17 and for ICT = 0.026.

	PHES		ICT	
	MHE+	MHE–	MHE+	MHE–
With decompensation	6 (37%)	7 (18%)	10 (38%)	3 (11%)
Without decompensation	10 (63%)	31 (82%)	16 (62%)	25 (89%)

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
