# Peer review of "Clinical Usefulness of the Inhibitory Control Test (ICT) in the Diagnosis of Minimal Hepatic Encephalopathy"

_ijerph, 2020, doi:10.3390/ijerph17103645_

Round 1

Reviewer 1 Report

The Authors of the paper studied the prevalence, risk factors and predictive value of minimal hepatic encephalopathy (MHE) and the usefulness of the inhibitory control test (ICT) in the diagnosis. This paper is a significant contribution to the scientific discussion about diagnosing MHE (usefulness of ICT). The limitation of the study is the small number of groups: MHE+ (n=21) and MHE- (n=49).   I suggest the revisions:

1) In: 2.2.3. Statistical analysis - Why were statistics processed by using two types of software: Statistica 11.0 and GraphPad Prism 5.0 ?

2) In discussion: The study carried out on a Poland's population; Using term ''European population'' is too wide.

2) In conclusion (or in disussion), the study's limitations must be mentioned.

Author Response

1) In: 2.3. Statistical analysis- Why were statistics processed by using two types of software: Statistica 11.0 and GraphPad Prism 5.0 ?

Answer: Statistica 11.0 was used for statistical analyses, and GraphPad Prism 5.0 was used to create graphs. Requested information was amended in the materials and methods section.

2) In discussion: The study carried out on a Poland's population; Using term ''European population''.

Answer: The manuscript was corrected according to reviewer suggestion and “European population was changed to Polish population.

3) In conclusion (or in disussion), the study's limitations must be mentioned.

Answer: Limitations of the study were included in the Discussions sections. The limitations included relatively small number of participants, especially in MHE+ groups but also quite significant proportion of patients lost to follow-up.

Reviewer 2 Report

Stawicka et al evaluated the use of the inhibitory control test vs. psychometric hepatic encephalopathy score in the diagnosis of minimal hepatic encephalopathy using 70 patients with cirrhosis and 56 controls. The authors found that ICT was somewhat useful in diagnosing MHE compared to the gold standard PHES (sensitivity 65-70%, specificity 57-67%). Presence of MHE was associated with more rapid increase in MELD. I have no major concerns regarding the analyses. I have several comments detailed below. Comments: 1. The test performance of ICT was somewhat worse in this study than what has been previously reported (see PMID 29533396 for a systematic review on this topic). The authors referenced a few other papers on ICT but I think further discussion is warranted. 2. How was the PHES cutoff of -5 chosen? Would provide a reference. 3. Would state whether any controls had abnormal PHES or ICT. 4. The authors state that MELD increased by 8.7 in PHES+ and 4.4 in PHES- patients, but 7.2 in ICT+ and 3.9 in ICT- patients. If all patients underwent both PHES and ICT, then why are the MELD increases higher in the PHES+/- groups? 5. The changes in MELD are quite large given that mean follow-up was only 17 months. Why was this the case? If this is due to a small number of outliers, recommend showing median rather than mean. 6. Table 2: would state whether longitudinal changes in decompensation status were statistically different based on MHE status. 7. Exclusion criteria: would specify which medications are considered “psychoactive agents”. I assume this refers to narcotics and benzodiazepines, but would tricyclic antidepressants, antipsychotics, gabapentinoids, etc. be included? Also, I believe that abnormal MMSE was used to identify neurologic disease but not necessarily the other exclusion criteria? If so, would say so explicitly. Minor comments: 1. The abbreviation PHES is not defined in the abstract.

Author Response

1) Interpretation of results and objective comparison between populations can be difficult due to previously adopted norms and cut-off point for the diagnosis of MHE by using PHES (e.g., Z-scores ≤ − 4 or ≤ − 5). In the our study a cut-off point ≤ −5 has been chosen, which is more conservative approach, but naturally clinical usefulness of ICT results may differ significantly compared to studies in which the cut off point was ≤ − 4. In addition, choosing a control group - in terms of age, sex and their effectiveness can cause significant differences in the interpretation of further results. This was discussed in the discussion section.

2) The text was supplemented.

3) In control group no subject reached cut-off of ≤ −5 in PHES (three subjects 6%) and one subject had % of correct responses in ICT <90.3%

4) and 5) Data has been checked again and recorded in the manuscript as median as requested by reviewer.  Naturally, their expression as mean or median did not change statistical significance of differences. The large variation MELD dynamics resulted mainly from severe deterioration of several patients due to the infections.

6) Statistical significances of differences were added to Table 2.

7) Exclusion criteria was the medications such as narcotics, benzodiazepines, tricyclic antidepressants, antipsychotics and gabapentinoids. Furthermore subjects with hepatocellular carcinoma or other malignancies were excluded. 
MMSE was performed by all patients regardless of disease symptoms. The test was an independent exclusion criterion.

Minor comments: 1. The text was supplemented.